# A Novel Graph and Safety Potential Field Theory-Based Vehicle Platoon Formation and Optimization Method

Linheng Li, Jing Gan , Xu Qu *, Peipei Mao, Ziwei Yi and Bin Ran

Collaborative Innovation Center of Modern Urban Traffic Technologies, School of Transportation, Southeast University, Nanjing 211189, China; leelinheng@seu.edu.cn (L.L.); jinggan@seu.edu.cn (J.G.); maopeipei@seu.edu.cn (P.M.); yiziwei@seu.edu.cn (Z.Y.); bran@seu.edu.cn (B.R.)
* Correspondence: quxu@seu.edu.cn; Tel.: +86-180-5209-9394

**Abstract:** Platooning is considered to be a very effective method for improving traffic efficiency, traffic safety and fuel economy under the connected and automated environment. The prerequisite for realizing these advantages is how to form a platoon without any collisions and how to maintain and optimize the car-following behavior after platoon formation. However, most of the existing studies focus on the platoon configuration and information transmission method, while only a few attempt to address the issue of platoon formation and optimization methods. To this end, this study proposes a novel platoon formation and optimization model combining graph theory and safety potential field (G-SPF) theory for connected and automated vehicles (CAVs) under different vehicle distributions. Compared to previous studies, we innovatively incorporate the concept of the safety potential field to better describe the actual driving risk of vehicles and ensure their absolute safety. A four-step platoon formation and optimization strategy is developed to achieve platoon preliminary formation and platoon driving optimization control. Three traffic scenarios with different CAVs distributions are designed to verify the effectiveness of our proposed platoon formation method based on G-SPF theory, and the simulation results indicate that a collision-free platoon can be formed in a short time. Additionally, the G-SPF-based platoon driving optimization control method is demonstrated by comparing it with two typical control strategies. Compared with the constant spacing and constant time headway control strategies, the simulation results show that our proposed method can improve the traffic capacity by approximately 48.8% and 26.6%, respectively.

**Keywords:** platoon formation; safety potential field; platoon control strategy; connected and autonomous vehicles

## 1. Introduction

The advancement in vehicular and communication technology facilitates vehicles to organize into a group of units with small inter-vehicle spacing commonly known as a vehicular platoon. In the connected and autonomous vehicles (CAVs) environment, the vehicle platoon is considered as a promising solution for enhancing road safety, improving road traffic efficiency and reducing fuel consumption in recent studies [1–5]. In addition, many scholars believe that CAVs have the potential to be applied in many scenarios in the future, such as car sharing systems [6–8], urban transport systems [9] and public transport systems [10,11]. Therefore, with the increase in the market penetration of CAVs, the research on the formation method and driving optimization of vehicle platoons will continue to be a research hotspot. The design of the formation progress includes the geometric configuration of the platoon, the cruising speed of the platoon and the mutual distance between vehicles in the platoon [12]. In the actual traffic system, the different intelligence levels of vehicles and the topological structure of the information flow between vehicles are very important factors that affect the formation of the platoons. The actual traffic system is a complex system composed of multiple traffic factors. Therefore, it is necessary to

comprehensively consider the actual road environment information (e.g., vehicle information, road information) in the formation process of the platoon. Inadequate analysis of the topological structure of information flow in the platoon and a single consideration of traffic environment variables will usually lead to inefficiency of the platoon in terms of traffic efficiency. A graph and safety potential field (G-SPF)-based vehicle platoon formation model is expected to fill this research gap. The purpose of this paper is to investigate the effectiveness potential of vehicle platoon formation using graph and safety potential field theories, along with a driving optimization control strategy on the basis of safety potential field theory.

The novelty and main contribution of this study lie in the combination of graph theory and potential field theory to achieve platoon formation and driving optimization that can be further implemented in the CAVs control system in the near future. This method is new to the field of platoon control. Compared with previous studies, we innovatively incorporate the concept of the safety potential field to better describe the actual driving risk of vehicles and ensure their absolute safety under the CAVs environment. In addition, our proposed method outperforms the traditional platoon control methods (constant spacing and constant headway) in terms of traffic efficiency.

The rest of this paper is organized as follows. A brief review of the existing related studies is introduced in Section 2. Section 3 presents the modeling process of the G-SPF platoon formation method. Numerical experiments under different initial states are performed in Section 4. Finally, the conclusions are summarized in Section 5.

## 2. Literature Review

Although a platoon formed by the aggregation of multiple vehicles can bring a certain improvement effect to the traffic flow, these improvements are assuming that the vehicle has completed the platoon forming process. Most existing research has focused on the control and optimization of vehicle platoons which have already completed their formation. However, how to form a platoon from randomly spaced vehicles has not yet been adequately addressed. At present, most of the research on this issue has been conducted from the perspective of cybernetics. The vehicles in the platoon are regarded as intelligent units, and the dynamic platoon system is coupled through the cooperative control of the intelligent units. A recent study by Kamel et al. [13] pointed out that the formation control strategy of agents can be divided into five types based on different research methods: the virtual structure strategy, the behavior-based strategy, the leader–follower strategy, the graph-based strategy and the artificial potential field strategy. According to the characteristics of each strategy, Table 1 summarizes the functions that each strategy can achieve during the formation process.

From Table 1, it can be found that the graph-based strategy and artificial potential field strategy have the most extensive abilities. These two strategies can be well applied to the formation control of intelligent agents. In particular, the graph-based strategy can take into account the topology structure between agents in the formation process. However, due to the kinematic limitations of vehicles and the complicated traffic environment of actual roads, there are essential differences between agents and vehicles. Therefore, the existing agent formation control strategies cannot be directly applied to vehicle platoons. Bang and Ahn described the connection between CAVs within a platoon as a spring-mass-damper system [14]. They completed the description of the concept of a platoon through the construction of this system. Taking the platoon velocity and space distance as optimization objects, Heinovski and Dressler constructed an optimization algorithm for platoon formation and verified it through simulations [15]. Ye et al. found that, in practice, a reasonable optimization of the number of vehicles in the platoon and the speed of the platoon can greatly improve the efficiency of road traffic and reduce fuel consumption [16].

**Table 1.** Summary of different platoon formation control strategies.

| Strategy | References | Realizable Functions | | | |
|---|---|---|---|---|---|
| | | Formation Shape Generation | Formation Trajectory Tracking | Formation Reconfiguration | Task Assignment |
| Virtual structure | [17,18] | | √ | | |
| Behavior-based | [19,20] | √ | √ | | √ |
| Leader–follower | [21,22] | | √ | | |
| Graph-based | [23,24] | √ | √ | √ | √ |
| Artificial potential field | [25,26] | √ | √ | √ | |

All of the above methods lack the consideration of complex traffic environment factors. The road traffic environment is composed of people, vehicles, roads and other traffic infrastructures. The layout of road signs and markings and the design of the road geometry will greatly affect the actual operation of vehicles because the impact of the traffic environment on vehicle travel is difficult to quantify with a specific indicator. In recent years, a novel research method for driving safety has been proposed [27–29]. In our previous study, we proposed a potential field model to better describe the driving behavior of vehicles under the CAVs environment [30]. The results indicate that the potential field model can well describe the driving risks constituted by the complex traffic environment and can be used to evaluate the potential driving risks in actual traffic scenarios. In this paper, we will further expand the previous research and focus on exploring the process of forming a platoon when multiple CAVs are in different initial states (different position and speed states), and an innovative combination of graph theory and safety potential field theory is developed to identify the geometric configuration and characterize the driving risk under complex traffic environments during the platoon formation process.

## 3. Methodology

In this section, the platoon formation model based on graph theory and safety potential field theory is employed to describe the topologies of a vehicle platoon and estimate the impact of the complex traffic environment variables on the performance of the platoon.

### 3.1. Platoon Topology Control Based on Graph Theory and Safety Potential Filed Theory

Figure 1 depicts the information transfer topology diagram in a vehicle platoon, including a leading vehicle (noted as the platoon leader and marked in color) and other following vehicles (noted as the follower vehicles, and marked sequentially from 1 to *i*). Communication technologies such as vehicle to vehicle (V2V)/vehicle to infrastructure (V2I) can be applied to realize the construction of the information flow topology, which can help CAVs to obtain the motion status information of surrounding vehicles. The arrows in Figure 1 represent the topological structure of information transmission between the vehicles in the platoon. These CAVs are capable of using this information to correct the distance (longitudinal and lateral) to surrounding vehicles and ultimately achieve formation control over the platoon.

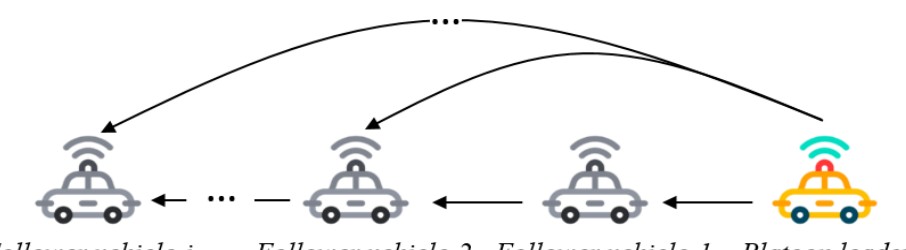

**Figure 1.** The information transfer topology diagram in a vehicle platoon.

The vehicle platoon in Figure 1 can be described by graph-based theory [31,32]. In graph theory, a directed graph has the expre*ssion G = (V\*, E), where V\* and E represent the vertex set (V\* = {v\*_i, i = 1 ··· n}) and the edge set (E ⊆ V\* × V\*, E = {e_k, k = 1 ··· n}) in* the directed graph, respectively. Every vertex represents an individual vehicle ($v_i^*$) and each edge represents a potential communication link ($e_k$). The directed edge implies the direction of communication transmission. For example, if the communication is transmitted from $v_i^*$ to $v_j^*$, then $v_i^*$ is defined as the parent node and $v_j^*$ is defined as the child node. Then, the directed edge has the expression $e_k = \left(v_i^*, v_j^*\right)$. Otherwise, the expression of $e_k$ will be rewritten as $e_k = \left(v_j^*, v_i^*\right)$ if the communication is transmitted from $v_j^*$ to $v_i^*$. The adjacency matrix $A = \left[a_{ij}\right]$ is a commonly used representation of a graph structure. It uses a digital square matrix to record whether there are edges connected between the vertexes. In a directed graph, adjacent matrices are not necessarily symmetric. Since the element in the *i*-th row and the *j*-th column of the adjacent matrix is 1, it means that there is an edge from vertex $v_i^*$ to vertex $v_j^*$, and there is not necessarily an edge from vertex $v_j^*$ to vertex $v_i^*$ at this time. Meanwhile, the incidence matrix $I = \left[I_{ij}\right]$ is a matrix that shows the relationship between two classes of objects: the vertex ($v^*$) and the edge ($e$). The values of the elements in the incidence matrix I should follow the rules below: (1) if the edge *e* is directed to the vertex *v*, then the value $I_{ij}(v^*, e) = -1$; (2) if the edge *e* is directed from the vertex $v^*$, then the value $I_{ij}(v^*, e) = 1$; (3) otherwise, the value $I_{ij}(v^*, e) = 0$ is used. The degree matrix $D = [d_i]$ contains information about the degree of each vertex. In a directed graph, the term degree may refer either to indegree (the number of incoming edges at each vertex) or outdegree (the number of outgoing edges at each vertex). The degree matrix D is used together with the adjacency matrix A to construct the Laplacian matrix $\mathcal{L}$ of a graph. The expression of the Laplacian matrix $\mathcal{L}$ can be shown as follows:

$$\mathcal{L} = D - A. \tag{1}$$

Assume that the importance of each edge is consistent with the degree matrix, with reference to the definition of the incidence matrix I, the Laplacian matrix $\mathcal{L}$ can also be expressed in the following variants [33]:

$$\mathcal{L} = I \cdot D \cdot I^T. \tag{2}$$

After the above analysis, it can be concluded that in our research, the vertexes (*V*) and edges (*E*) correspond to the controlled CAVs and inter-vehicle communication links for information sharing, respectively. The following theorem is used to offer a stable solution to the platoon formation problem based on graph theory.

**Theorem 1.** *Suppose that the motion control function of the individual vehicle can be expressed by $f(\cdot)$. $f(\cdot)$ can be an N-dimensional vector, and $f_i$ is the value of the function $f(\cdot)$ at the vertex $v_i$ in the graph. It is assumed that under the CAVs environment, the vehicles in a platoon have a specific communication topology through V2V (vehicle to vehicle)/V2X (vehicle to Everything). Changes in the motion of every following vehicle can be directly affected by the state of the leading vehicle in the platoon.*

Then, we have

(i).　$f(\cdot) = \begin{bmatrix} \dot{s} \\ \vdots \\ l \end{bmatrix}$.

(ii).　$\Delta f(\cdot) = \mathcal{L} \cdot f(\cdot)$.

In particular, under certain conditions, the vehicles driving on the road can form a platoon based on graph theory. This means that the longitudinal distance *s* and the lateral distance *l* can converge to the desired value through the Laplacian matrix $\mathcal{L}$.

**Proof.** *If there is a disturbance in the motion state $f$ of any vertex $j$ ($v_j$), the gain change caused by this disturbance to vertex $j$ connected to vertex $i$ ($v_i$) can be expressed as follows (where, $j \in N_i$, $N_i$ represents the first-order neighborhood node of the vertex $i$ and considering that the weights of the edges in the graph are the same, equal to 1):*

$$\Delta f_i = \sum_{j \in N_i} (f_i - f_j). \tag{3}$$

*If edge $e_{ij}$ has the corresponding weight $w_{ij}$, then*

$$\Delta f_i = \sum_{j \in N_i} w_{ij}(f_i - f_j). \tag{4}$$

*When $w_{ij} = 0$, there is no connection between vertex $i$ and vertex $j$, that is to say, there is no communication between $v_i$ and $v_j$. Then, Equation (4) can be simplified as*

$$
\begin{aligned}
\Delta f_i &= \sum_{j \in N} w_{ij}(f_i - f_j) \\
&= \sum_{j \in N} w_{ij} f_i - \sum_{j \in N} w_{ij} f_j \\
&= d_i f_i - W_i f.
\end{aligned}
\tag{5}
$$

*Consequently, for all of the N vertexes, we have*

$$
\begin{aligned}
\Delta f &= \begin{pmatrix} d_1 & \cdots & 0 \\ \vdots & \ddots & \vdots \\ 0 & \cdots & d_N \end{pmatrix} f - \begin{pmatrix} W_1 \\ \vdots \\ W_N \end{pmatrix} f \\
&= diag(d_i)f - Wf \\
&= (D - W)f \\
&= \mathcal{L}f.
\end{aligned}
\tag{6}
$$

*The proof is completed.* □

**Remark 1.** *According to Theorem 1, in a vehicle platoon composed of CAVs, if the communication topology between CAVs is defined, then a graph based on the topology can be obtained, and the Laplacian matrix of the graph can be applied to achieve platoon formation with the desired distance and speed. In particular, according to whether there is a communication connection between the vehicles, the vehicles connected to each other can correct the difference between the actual distance and the expected distance through real-time communication. Thus, the longitudinal and the lateral distances between CAVs can be kept in the safe range, and the velocity of each CAV can track the desired velocity.*

To describe the scenario in Figure 1, a typical graph-based control protocol is shown as follows:

$$\Delta f = \mathcal{L} \begin{bmatrix} \dot{s} \\ \dot{l} \end{bmatrix} = \mathcal{L} \begin{bmatrix} f(d_s - s_a) \\ f(d_l - l_a) \end{bmatrix}, \tag{7}$$

where $d_s$ and $d_l$ are the desired longitudinal distance and lateral distance between two vehicles which are connected by a mutual communication connection, respectively. Correspondingly, $s_a$ and $l_a$ are the actual longitudinal distance and lateral distance between two communication connection vehicles, respectively. $f(d_s - s_a)$ and $f(d_l - l_a)$ express the effect function of the deviation between the desired distance and the actual distance on the vehicle acceleration. For a more detailed explanation of this function, please refer to the studies of [32,34].

The control method mentioned above is conducive to the rapid formation of the vehicular platoon but only considers two factors in terms of the space distance and vehicle communication method. In many studies [15,32], similar ideas have been used to ensure

the efficiency of platoon formation. However, it is worth noting that the movement status (the value of the acceleration, the direction of the velocity, the value of the steering angle, etc.) of each vehicle in the platoon and the information of the road environment (signal period of the intersection, the speed limit of the road, etc.) will affect the driving of the platoon. Therefore, it is essential to summarize and analyze the impact of those various factors on the forming of the vehicle platoon. For this, we developed a platoon formation control model and a platoon optimization control model based on graph theory and safety potential field theory, and more details are discussed in the following sections.

### 3.1.1. Platoon Formation Control Model

The formation of a platoon will definitely involve the lane changing process, and vehicles need to consider the complex road environment risk (e.g., surrounding vehicle information, road line information) to make motion decisions. It is necessary to ensure the absolute safety of the vehicles during the formation of the platoon. The safety potential field of a complex road environment can be divided into three categories: the lane marking potential field $E_L$, the road boundary potential field $E_B$ and the vehicle potential field $E_V$ [30]. These three kinds of potential field can be expressed as

$$
\begin{aligned}
E_v &= M_i\lambda\frac{e^{-\beta_1 a\cos\theta}}{|k'|^{\zeta}}\cdot\frac{k'}{|k'|} \\
&= m_i\times\left(1.566\times10^{-14}v^{6.687}+0.3345\right)\lambda\frac{e^{-\beta_1 a\cos\theta}}{\left\{\sqrt{\left[(x-x_0)\frac{\tau}{e^{\Delta v}}\right]^2+[(y-y_0)\tau]^2}\right\}^{\zeta}}\cdot\frac{k'}{|k'|},
\end{aligned}
\tag{8}
$$

$$
\begin{aligned}
E_L &= \sum_{i,j=1}^{n}A_ie^{\left(-\frac{|d_{Aj}^L|}{2\sigma^2}\right)}\cdot\frac{d_{Aj}^L}{\left|d_{Aj}^L\right|}, \\
d_{Aj}^L &= y_{l,j}-y_A,
\end{aligned}
\tag{9}
$$

$$
\begin{aligned}
E_B &= \tfrac{1}{2}\eta\left(\frac{1}{\left|d_{left}^B\right|}\right)^2\cdot\frac{d_{left}^B}{\left|d_{left}^B\right|}+\tfrac{1}{2}\eta\left(\frac{1}{\left|d_{right}^B\right|}\right)^2\cdot\frac{d_{right}^B}{\left|d_{right}^B\right|}, \\
d_{left}^B &= \left(y_{left}-y_A\right),d_{right}^B=\left(y_{right}-y_A\right),
\end{aligned}
\tag{10}
$$

where $\lambda$, $\beta_1$ and $\zeta$ are determined coefficients, $k'$ is the pseudo-distance, $M_i$ denotes the equivalent mass of object vehicle $i$, $\theta$ is the clockwise angle formed by any point around the object vehicle and the mass center of the object vehicle with the motion direction of the vehicle and $a$ is the acceleration of the current motion state of the object vehicle. $A_i$ represents the field intensity coefficient of different types of lane marking, which determines the maximum value of the lane marking potential field intensity. For example, the value of double amber lines $A_2$ is much larger than that of lane-dividing lines $A_1$. $y_A$ is the ordinate value of point A in the road and $y_{l,j}$ represents the position coordinates of the $j\_th$ lane marking along the $Y$-axis; $d_{Aj}^L$ represents the distance vector from point A to the $j\_th$ lane marking; $\sigma$ determines the velocity at which the potential field increases or decreases as the vehicle approaches or moves away from the lane marking. $d_{left}^B$ and $d_{right}^B$ represent the distance vector from point A to the left and right road boundaries, respectively. $y_{left}$ and $y_{right}$ are the position coordinates of the road boundary on the left and right sides, respectively. Finally, $\eta$ is the coefficient of the road boundary potential field.

Figure 2 displays the road boundary potential field (black dotted lines), the lane marking potential field (green dotted lines), the road vehicle potential field (red dotted lines) and the total potential field (blue solid line). The strength of the potential field at a certain point on the road represents the level of the potential traffic safety risk at that point. It can be found that the strength of the potential field reaches the maximum at the road boundary. The strength of the potential field is also relatively high at the position of the vehicle and the double yellow line. This characterization of the driving risk degree is completely consistent with the safety risks in the actual driving process of the vehicle. In

our previous studies [29,35], the vehicle potential field model was proven to accurately describe the driving risk of vehicles under different motion states in the CAVs environment.

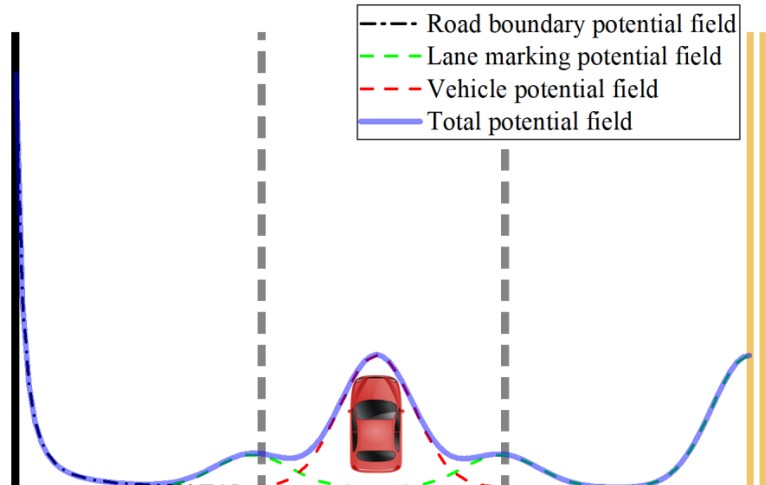

**Figure 2.** Cross-sectional map of the potential field.

Since the constraints of the complex road environment are not taken into account during vehicle driving, as proved in Equation (7), in this paper, we combine the potential field model that characterizes vehicle driving safety with the graph theory model, by introducing a velocity-based slack variable based on the potential field and a function of the road environment total potential field shown as the blue line in Figure 3. The function of the road environment potential field is employed to describe the control constraints of the road environment on vehicles. Thus, we can obtain

$$\Delta f = \mathcal{L} \left[ \begin{array}{c} \dot{s} \\ \dot{l} \end{array} \right] = \mathcal{L} \left[ \begin{array}{c} f(d_s - s_a) \\ f(d_l - l_a) \cdot g(E_e) \end{array} \right],$$

$$g(E_L) = \left\{ \begin{array}{cc} 1 & E_e \leq E_0 \\ 0 & E_e > E_0 \end{array} \right., \tag{11}$$

where $E_e$ and $E_0$ represent the potential field intensity of the actual road environment and dotted road line, respectively.

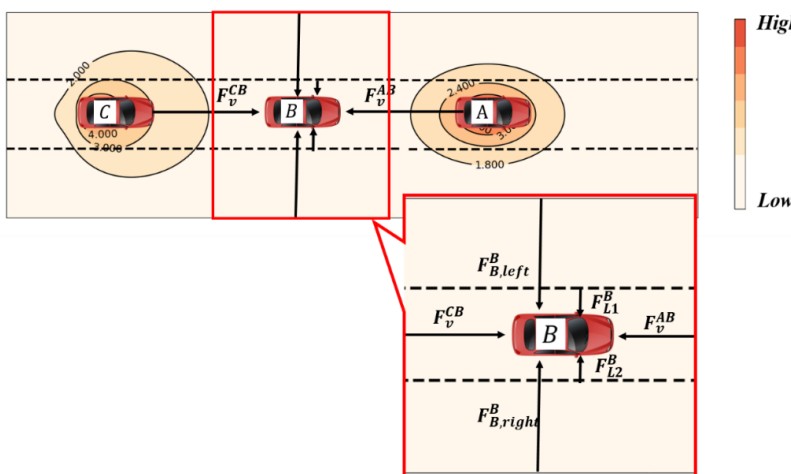

**Figure 3.** Different types of field forces on the vehicle B.

3.1.2. Platoon Formation Control Model

After the formation of the platoon, the optimal car-following control of vehicles in the platoon is key to improving traffic efficiency and avoiding string instability. In this paper, we introduce a slack variable $SV$ based on the safety potential model to the effect function $f$ to address the problem of optimal car-following control, as shown in Equation (8). The slack variable is a comprehensive velocity gain of the vehicle caused by factors such as the vehicle state and traffic environment. $SV_s$ and $SV_l$ are the components of the slack variable in longitudinal and lateral directions, respectively.

$$\Delta f = \mathcal{L} \begin{bmatrix} \dot{s} \\ \dot{l} \end{bmatrix} = \mathcal{L} \left( \begin{bmatrix} f(d_s - s_a) \\ f(d_l - l_a) \end{bmatrix} + \begin{bmatrix} SV_s \\ SV_l \end{bmatrix} \right). \tag{12}$$

In our previous studies, the velocity control of any vehicle $i$ in a platoon according to the safety potential field theory was characterized as

$$\dot{v}_i(t+1) \quad = a_{free} + a_{ji}$$
$$= a_{max} \tanh \left[ \delta \cdot \left( v_0^{(\alpha)} - v_i(t) \right) \right] - e^{\beta_2 \cdot v_i(t) \cdot \cos \phi} \cdot M_j \cdot \lambda \cdot \frac{e^{-\beta_1 \cdot a_j(t) \cdot \cos \theta}}{|k'|^\zeta} \cdot \frac{k'}{|k'|}, \tag{13}$$

where $a_{max}$ represents the maximum allowable acceleration of the vehicle, $a_{free}$ represents the acceleration of vehicle $i$ when the distance between vehicle $i$ and the leading vehicle (vehicle $j$) is long enough, $v_0^{(\alpha)}$ is the desired velocity of vehicle $i$, $v_i(t)$ represents the velocity of vehicle $i$ at time $t$, $M_j$ denotes the equivalent mass of object vehicle $j$, $\beta_2$ is an undetermined coefficient and $\phi$ refers to the angle between the direction of the vehicle velocity and the $X$-axis (counter-clockwise).

Note that the velocity-based slack variable $SV$ reflects the feedback gain of the vehicle's driving environment on the vehicle velocity. In this paper, $SV$ is defined as a variable that changes the vehicle's motion state under the influence of the safety potential field, which is directly reflected in the speed change of the vehicle. Thus, the definition of the $SV$ based on safety potential field theory is actually modeling the impact of the complex driving environment on the vehicle. The velocity-based slack variable in the lateral direction $SV_s$ can be defined as the lateral velocity control expression in Equation (13) to achieve the speed optimization of the individual vehicles in the platoon.

For the expression of the velocity-based slack variable in the longitudinal direction $SV_l$, the lane marking potential field and the road boundary potential field are employed to describe the constraints of the lane marking and road boundary to ensure that the vehicle does not produce a deviation in the longitudinal direction and thus to enable the vehicle to keep driving in the center of the lane.

Then, as shown in Figure 3, the field forces generated by these two types of safety potential field will reach the longitudinal motion constraint of the vehicle. Similar to the field force analysis formed by the vehicle potential field, we can also derive the respective field force expressions based on the safe potential field expressions shown in Equations (9) and (10). Then, we can obtain the function expression of $SV_l$ based on the field force expressions of the road boundaries and road lines [29]. Finally, the comprehensive expression of the velocity-based slack variable $SV$ can be shown in Equation (14).

$$SV = \begin{bmatrix} SV_s \\ SV_l \end{bmatrix} = \begin{bmatrix} a_{max} \tanh \left[ \delta \cdot \left( v_0^{(\alpha)} - v_i(t) \right) \right] - e^{\beta_2 \cdot v_i(t) \cdot \cos \phi} \cdot M_j \cdot \lambda \cdot \frac{e^{-\beta_1 \cdot a_j(t) \cdot \cos \theta}}{|k'|^\zeta} \cdot \frac{k'}{|k'|} \\ \sum_{k=1}^{n} e^{\beta_2 v_i(t) \cdot \sin \phi} \cdot A_i \cdot e^{\left( -\frac{|d_{Lk}^i|}{2\sigma^2} \right)} + e^{\beta_2 v_i(t) \cdot \sin \phi} \cdot \frac{1}{2}\eta \left( \left( \frac{1}{|d_{right}^i|} \right)^2 + \left( \frac{1}{|d_{left}^i|} \right)^2 \right) \end{bmatrix} \tag{14}$$

3.2. Platoon Formation and Optimization Control Strategy

In order to achieve efficient vehicle formation operation, the control of the leading vehicle and the following vehicles is subsequently discussed to construct a complete

platoon formation control model. The entire control model based on graph theory and safety potential field theory consists of four steps: the identification of vehicles that can potentially form a platoon, the recognition of the leading vehicle in a platoon, the formation of a preliminary platoon and the implementation of an optimal driving strategy for vehicles in a platoon accounting for traffic efficiency and traffic safety. Figure 4 illustrates the flow chart of our proposed method. The variable t represents the time interval for each vehicle to judge whether there is a chance to form a platoon. The indicator of the end of the algorithm is the horizontal and vertical distances between the vehicles when they reach the desired distance.

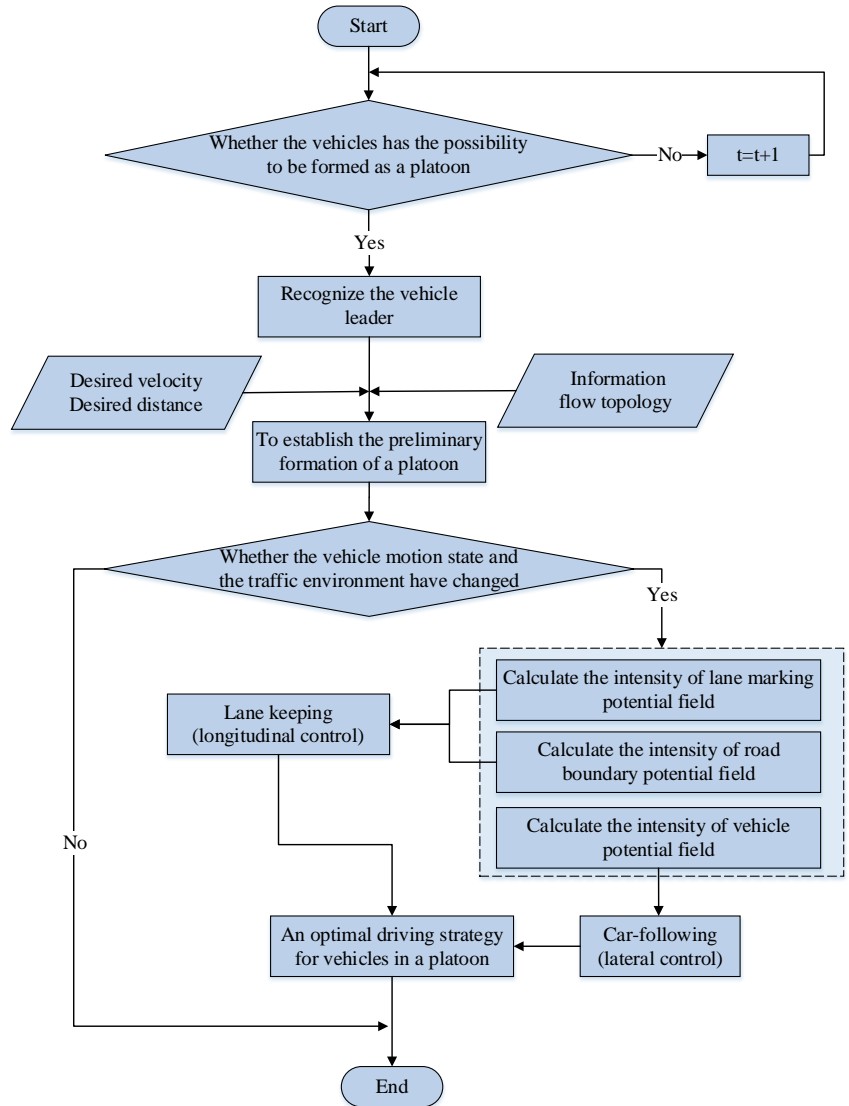

**Figure 4.** Flow chart of the platoon formation control strategy.

Step 1: The identification of vehicles that may form a platoon

The arrival of vehicles on the road is a random phenomenon, and the longitudinal and lateral distances between adjacent vehicles are uncertain in the initial stage. Therefore, some basic constraints should be established to identify those vehicles that may form a platoon. Two microscopic traffic indexes (i.e., vehicle headway and vehicle velocity) are commonly used to identify the vehicles which can form a platoon [4]. In this paper, we additionally use the transmission distance of the information flow as the third index to

describe the limitation of the communication distance between vehicles in a platoon. The constraints in Equation (15) should be satisfied with the vehicles in a platoon as follows:

$$\begin{cases} H_i \leq H_0 \\ V_i \leq V_0 \\ D_{l-i} \leq D_{\max} \\ V_i = |v_i - v_{i-1}| \end{cases} \tag{15}$$

where $H_i$ is the space headway ($m$) of two adjacent vehicles (vehicle $i$ and vehicle $i-1$) in the same platoon, $H_0$ is the critical value of the space headway ($m$) of two adjacent vehicles, $V_i$ is the absolute difference in velocity ($m/s$) between two adjacent vehicles (vehicle $i$ and vehicle $i-1$), $v_i$ and $v_{i-1}$ are the velocities of vehicle $i$ and vehicle $i-1$, respectively, $V_0$ is the critical value of the absolute difference in velocity ($m/s$) between two adjacent vehicles, which can ensure that the vehicle generates as little disturbance (from the sudden increase or decrease in vehicle velocity) as possible during the platoon formation process, $D_{l-i}$ is the distance ($m$) between the leading vehicle and the following vehicle $i$ and $D_{max}$ is the maximum transmission distance ($m$) that the leading vehicle can transmit in the information flow topology of a platoon.

According to the constraint requirements in Equation (15), we can make a preliminary judgment on whether vehicles have the possibility of forming a platoon.

Step 2: Identification and control of the platoon leader

According to the result of Step 1, if it can be determined that a group of vehicles can have the potential to form a platoon, then the leading vehicle of the platoon can be determined based on the natural spatial location of the vehicle. The first vehicle in the positive direction of the platoon is automatically recognized as the leading vehicle. For the special situation that some vehicles can form a platoon with the vehicles in front or with the vehicles behind, this article chooses the first vehicle in the driving direction as the leading vehicle selection criterion to form the first platoon. If the vehicles outside the communication range of platoon 1 meet the judgment condition of Step 1 with their following vehicle, this will automatically form a second platoon.

Step 3: The preliminary formation of a platoon

After exploring the identification and control of the leading vehicle, the preliminary formation of a platoon is constructed. The factors that need to be considered in the preliminary vehicle formation are the difference between the actual distance between the vehicles ($S_{i-1}(t) - S_i(t)$) and the desired distance ($d_s$) and the limitations of the vehicle's motion attributes (the value of the vehicle velocity ($v_i(t)$) cannot be greater than the value of the desired velocity ($v_0$), and the acceleration ($a_i(t)$) and deceleration ($b_i(t)$) of the vehicle cannot exceed the values of the maximum acceleration ($a_{max}$) and maximum deceleration ($b_{max}$) respectively). The specific expression is shown in Equation (16).

$$\begin{cases} S_{i-1}(t) - S_i(t) - d_s = 0 \\ |v_i(t)| \leq |v_0| \\ |a_i(t)| \leq |a_{\max}| \\ |b_i(t)| \leq |b_{\max}| \end{cases} \tag{16}$$

In addition, it is necessary to combine the information flow topology of the platoon, which means that not only the front and rear vehicles but also the vehicles with information interaction with the platoon leader need to be calibrated in the platoon. Therefore, it is necessary to combine the corresponding graph-based control protocol of Equation (11) to control the vehicle and finally establish a preliminary platoon configuration.

Note that if the vehicle's movement state does not change (the vehicle maintains a constant velocity in real time) and the traffic environment does not change (the road velocity limit value does not change, the road geometry does not change, etc.), the driving

strategy of the platoon does not need to change. However, once there is a motion change of any vehicle, it is necessary to perform Step 4 to optimize the design of the driving strategy.

Step 4: The optimal driving strategy for a platoon

Consistency in vehicle movement behavior was considered as the objective function to improve the traffic efficiency of the platoon in many previous studies (fixed space headway between each adjacent vehicles), which ignores the impact of dynamic changes in the motion status on optimization results. After the formation of the platoon, the control method shown in Equation (12) in Section 3.1.2 is applied to find the optimal driving strategy. The optimization is mainly reflected in the car-following performance in the lateral direction and lane-keeping performance in the longitudinal direction, and this method can adjust the movement behavior of the vehicles in the platoon through the dynamic change in the distribution of the safety potential field. Under such a control method, both driving safety and traffic efficiency can be improved for the platoon.

## 4. Numerical Experiments

In this section, we perform several numerical experiments to verify the effectiveness of the proposed control strategy for the platoon forming process and the optimization strategy after the formation of the platoon. The tool used in the experiments is based on the MATLAB2020, the computer processor used is the Intel Core i7-8700@3.20 GHz, the graphics card is the NVIDIA GeForce GTX 1050Ti, the memory is 16 GB and the memory of the VRAM is 128 MB. In this section, three scenarios with different initial states (the spatial distribution and velocity of vehicles) are discussed. In order to make a reasonable comparison, we provide the same model parameter values according to our previous research [30]. The parameters are listed in Table 2.

**Table 2.** Parameter values [30].

| Parameter | Values | Parameter | Values | Parameter | Values |
|-----------|--------|-----------|--------|-----------|--------|
| $A_1$ | 2.00 | $\lambda$ | 0.056 | $a_{max}$ | 4.001 |
| $A_2$ | 8.00 | $\alpha$ | 0.029 | $b_{max}$ | 3.00 |
| $\sigma$ | 1.22 | $\beta_1$ | −0.169 | $v_0^{(\alpha)}$ | 22.032 |
| $\eta$ | 3.00 | $\beta_2$ | 0.117 | $H_0$ | 5.00 |
| $\delta$ | 4.647 | $\zeta$ | 1.09 | $\zeta$ | 1.09 |

**Scenario I**: In this scenario, all the following vehicles are distributed in the left lane of the leading vehicle at the initial state and are controlled to gradually merge into the lane where the leading vehicle is located one by one by adjusting their own positions. The spatial distribution $\left[X^i, Y^i\right]$ and the velocity $\left[v_x^i, v_y^i\right]$ of each *i-th* CAV at the initial state were set as given in Equation (17). Note that we use positive and negative signs to distinguish the speed direction in the Y direction. The vehicle velocity is negative when turning right and positive when turning left.

$$\begin{bmatrix} X^0 & Y^0 \\ X^1 & Y^1 \\ X^2 & Y^2 \\ X^3 & Y^3 \\ X^4 & Y^4 \\ X^5 & Y^5 \end{bmatrix} = \begin{bmatrix} 50 & 2 \\ 40 & 8 \\ 31 & 10 \\ 20 & 8 \\ 15 & 10 \\ 0 & 11 \end{bmatrix} (m), \quad \begin{bmatrix} v_x^0 & v_y^0 \\ v_x^1 & v_y^1 \\ v_x^2 & v_y^2 \\ v_x^3 & v_y^3 \\ v_x^4 & v_y^4 \\ v_x^5 & v_y^5 \end{bmatrix} = \begin{bmatrix} 6.0 & 0.0 \\ 3.0 & -1.15 \\ 14.0 & -2.15 \\ 8.0 & 0.35 \\ 9.0 & -0.15 \\ 9.5 & -0.4 \end{bmatrix} (m/s). \quad (17)$$

The elements $X^i$ and $Y^i$ represent the lateral and longitudinal position coordinates of the *i-th* CAV in spatial coordinates, respectively. Similarly, $v_x^i$ and $v_y^i$ represent the lateral and longitudinal velocities of the *i-th* CAV, respectively. Index $i = 0$ indicates the platoon leader and $i = 1, 2, 3 \cdots$ indicates the following vehicle $i$ according to the position of the vehicle.

Figure 5 depicts the trajectories of the CAVs during the platoon formation process. The gray vehicle is identified as the vehicle out of the communication range in the first step, and thus it will not participate in platoon formation. Then, the red vehicle is identified as the leading vehicle based on the longitudinal position of the vehicle at the initial moment at the second step. The other vehicles are controlled through Step 3 to form a platoon. From Figure 5, it can be found that the leading vehicle keeps driving in its own lane without any deviation, and the following vehicles are gradually merged into the lane where the leading vehicle is located one by one by adjusting their own positions. Note that Figure 6a,b present the changes in the lateral and longitudinal velocities of CAVs during the platoon formation process, respectively. Figure 6a shows that the lateral velocities of the following vehicles can converge to the desired velocity (the velocity of the leading vehicle) and become stable in the end. Figure 6b shows that the lateral velocities of the following vehicles converge to zero when the platoon is formed, which means that all the vehicles are in the same lane and no longitudinal velocity component is generated. According to Figures 5 and 6, it can be verified that all CAVs in scenario I can complete the platoon formation process without any collisions by the method proposed in this paper.

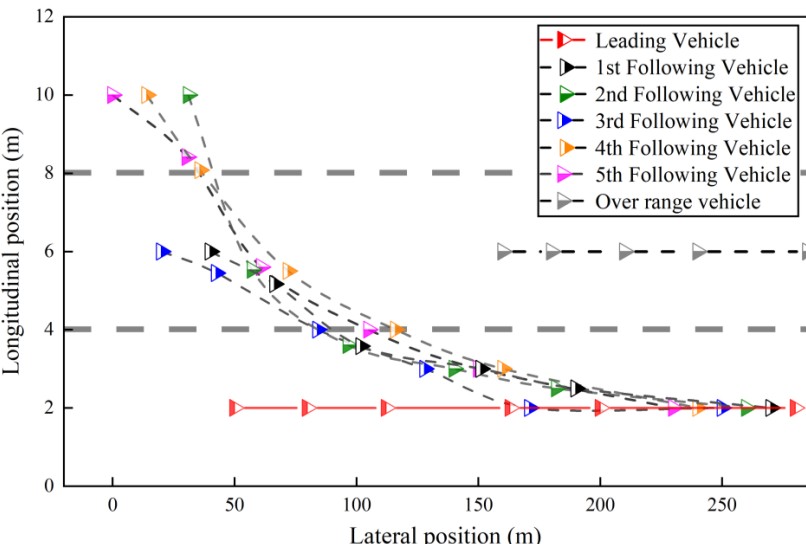

**Figure 5.** The trajectories of the connected and automated vehicles (CAVs) during the platoon formation process (scenario I).

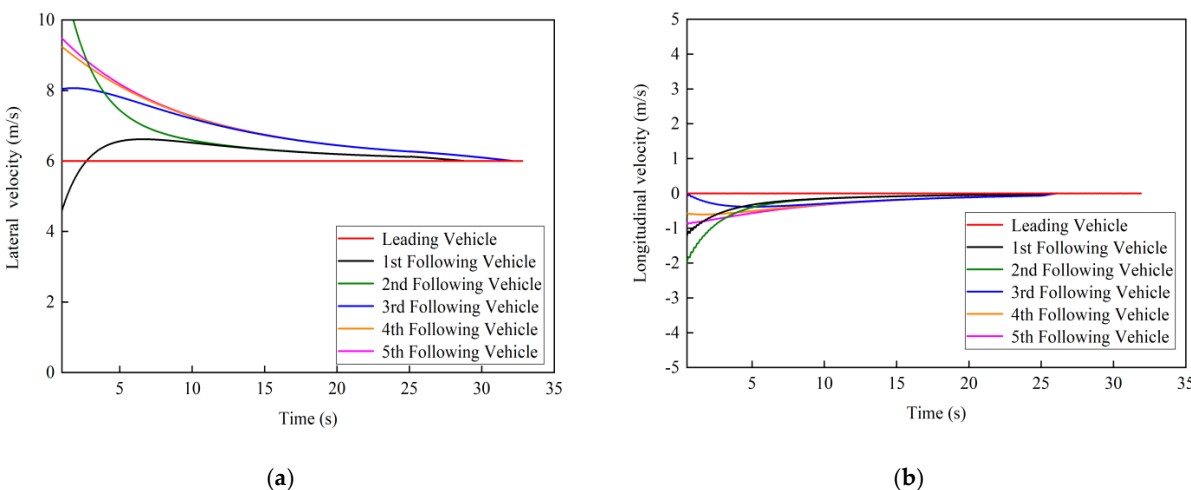

**Figure 6.** The velocities of the CAVs during the platoon formation process: (**a**) lateral velocity; (**b**) longitudinal velocity (scenario I).

**Scenario II**: In this scenario, all the following vehicles are distributed in the right lane of the leading vehicle at the initial state and are controlled to gradually merge into the lane where the leading vehicle is located one by one by adjusting their own positions. To show the advantages of the model in considering the complex road environment, in this scenario, we added double yellow lines on the road to simulate the platoon formation process in the merging area. The spatial distribution $\left[X^i, Y^i\right]$ and the velocity $\left[v_x^i, v_y^i\right]$ of each *i-th* CAV at the initial state were set as follows:

$$
\begin{bmatrix}
X^0 & Y^0 \\
X^1 & Y^1 \\
X^2 & Y^2 \\
X^3 & Y^3 \\
X^4 & Y^4 \\
X^5 & Y^5
\end{bmatrix}
=
\begin{bmatrix}
86 & 2 \\
60 & 6 \\
40 & 10 \\
38 & 6 \\
25 & 10 \\
5 & 10
\end{bmatrix}
(m),
\begin{bmatrix}
v_x^0 & v_y^0 \\
v_x^1 & v_y^1 \\
v_x^2 & v_y^2 \\
v_x^3 & v_y^3 \\
v_x^4 & v_y^4 \\
v_x^5 & v_y^5
\end{bmatrix}
=
\begin{bmatrix}
6.0 & 0.0 \\
8.0 & 0.0 \\
14.0 & 0.0 \\
8.5 & 0.0 \\
8.0 & 0.0 \\
9.5 & 0.0
\end{bmatrix}
(m/s).
\tag{18}
$$

Figures 7 and 8 present the trajectories and the lateral and longitudinal velocities of the CAVs during the platoon formation process, respectively. Similar to scenario I, all the vehicles except for the gray vehicle out of the communication range can be gradually merged into the same lane to form a platoon through our proposed control strategy in Section 4. Note that the first, second, third and fourth vehicles are controlled to maintain the car-following state in the double yellow line area where lane changing is prohibited. They start to change lanes when reaching the dotted line area and gradually merge into the lane of the leading vehicle. Compared with the previous study of platoon formation based on graph theory, consideration of the driving risk in the surrounding environment of the vehicle can better ensure traffic safety. This advantage comes from the accurate description of the safety risk of the driving environment by the safety potential field model.

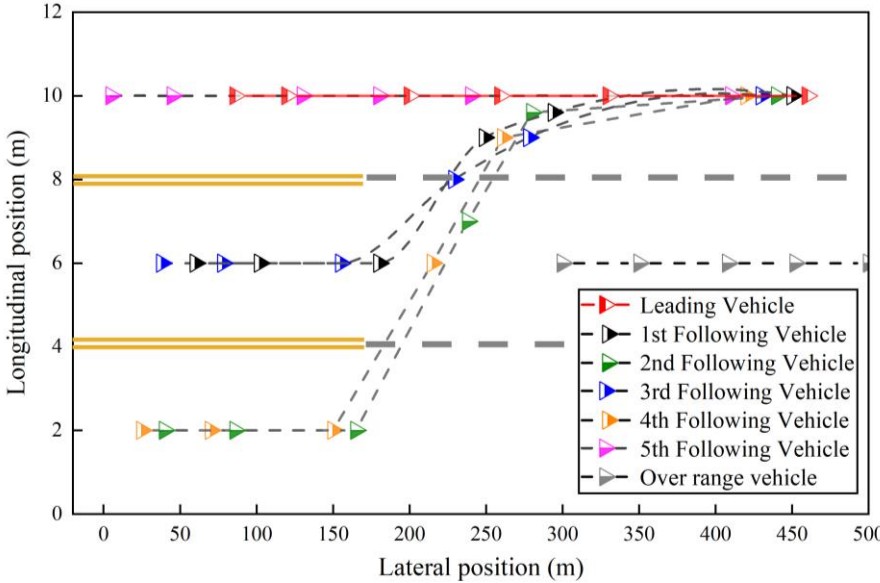

**Figure 7.** The trajectories of the CAVs during the platoon formation process (scenario II).

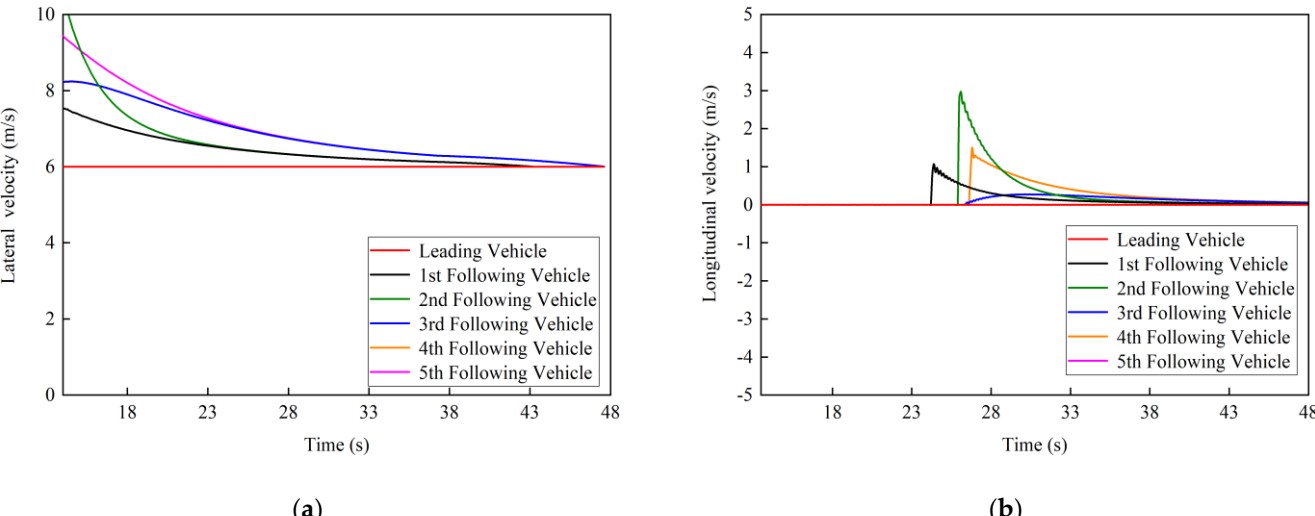

**Figure 8.** The velocities of the CAVs during the platoon formation process: (**a**) lateral velocity; (**b**) longitudinal velocity (scenario II).

**Scenario III**: In this scenario, the following vehicles are randomly distributed in the different lanes (left and right sides of the leading vehicle) at the initial state and are controlled to gradually merge into the lane where the leading vehicle is located one by one by adjusting their own positions. The spatial distribution $\left[X^i, Y^i\right]$ and the velocity $\left[v_x^i, v_y^i\right]$ of each *i-th* CAV at the initial state were set as follows:

$$\begin{bmatrix} X^0 & Y^0 \\ X^1 & Y^1 \\ X^2 & Y^2 \\ X^3 & Y^3 \\ X^4 & Y^4 \\ X^5 & Y^5 \end{bmatrix} = \begin{bmatrix} 70 & 6 \\ 60 & 6 \\ 48 & 2 \\ 40 & 10 \\ 22 & 2 \\ 5 & 10 \end{bmatrix} (m), \begin{bmatrix} v_x^0 & v_y^0 \\ v_x^1 & v_y^1 \\ v_x^2 & v_y^2 \\ v_x^3 & v_y^3 \\ v_x^4 & v_y^4 \\ v_x^5 & v_y^5 \end{bmatrix} = \begin{bmatrix} 6.0 & 0.0 \\ 4.3 & 0.0 \\ 7.8 & 2.0 \\ 6.4 & -1.0 \\ 9.15 & 1.0 \\ 12.15 & -1.0 \end{bmatrix} (m/s). \tag{19}$$

Figures 9 and 10 depict the trajectories and the lateral and longitudinal velocities of the CAVs during the platoon formation process, respectively. Similar to scenario I and scenario II, all the vehicles can also be controlled to gradually merge into the same lane to form a platoon through our proposed control strategy in Section 4.

The simulation results of scenarios I to III show that the platoon formation method proposed in this paper can be applied to the preliminary platoon formation of multiple CAVs in different initial states. A stable platoon is formed after approximately 20 s in all three scenarios. To perform a better analysis of our proposed optimal driving strategy for a platoon in Section 4, a certain external disturbance is applied to the leading vehicle of the platoon at 30 s, and the advantages of the optimal driving strategy in this paper are analyzed by observing the changes in the following vehicles in the platoon. The disturbance period lasts for 60 s and the simulation step is 0.1 s with the largest control acceleration and deceleration of 2 m/s$^2$ and $-2$ m/s$^2$, respectively.

To verify the advantages of our proposed potential field-based optimization method in Section 4, the constant spacing strategy in Ali (2015) and the constant time headway strategy in Viel et al. (2020) are applied for comparison and analysis. Figure 11 shows the comparison results in term of velocity, acceleration and gap distance under three different control strategies.

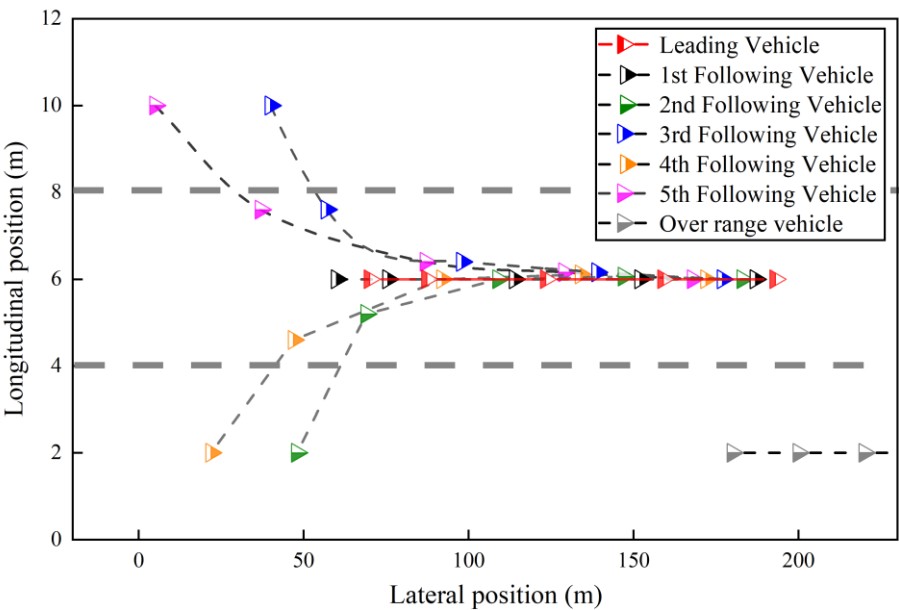

**Figure 9.** The trajectories of the CAVs during the platoon formation process (scenario III).

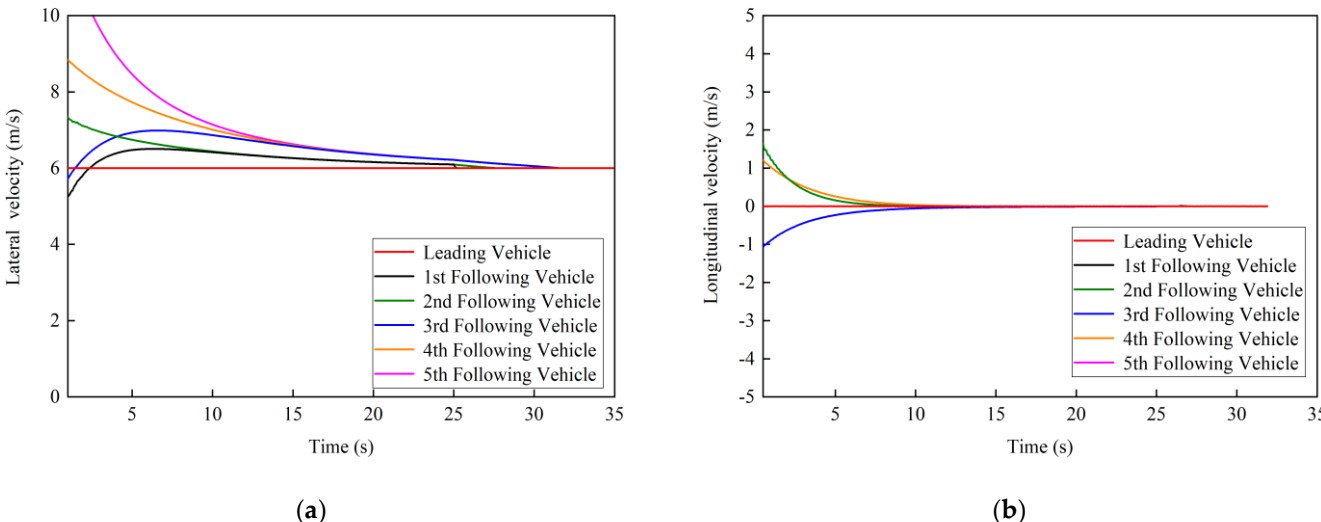

(**a**)                                                                 (**b**)

**Figure 10.** The velocities of the CAVs during the platoon formation process: (**a**) lateral velocity; (**b**) longitudinal velocity (scenario III).

Figure 11a–c show the changes in velocities of all vehicles in the platoon under three different control strategies. The velocity tracking effects of the constant time headway strategy and the potential field-based strategy are better, while the constant spacing strategy has a larger velocity deviation, with a maximum deviation of 1.42 m/s.

Figure 11d–f display the changes in accelerations of all vehicles in the platoon under three different control strategies. In the constant spacing strategy, the acceleration fluctuation changes the most, and the acceleration fluctuation of the constant time headway strategy has the best convergence effect. However, at the beginning of the disturbance, the acceleration fluctuates greatly.

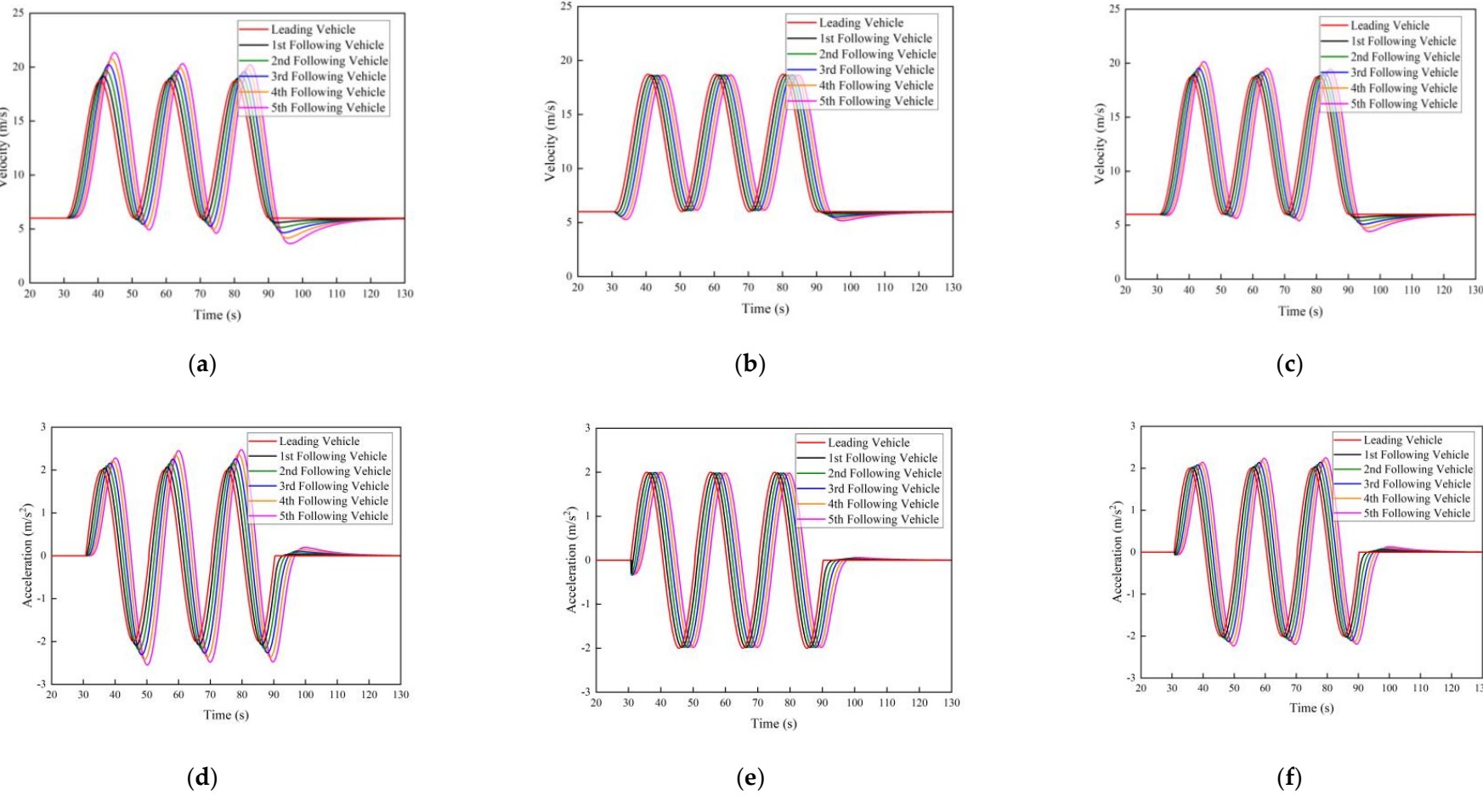

(a)  (b)  (c)

(d)  (e)  (f)

**Figure 11.** *Cont.*

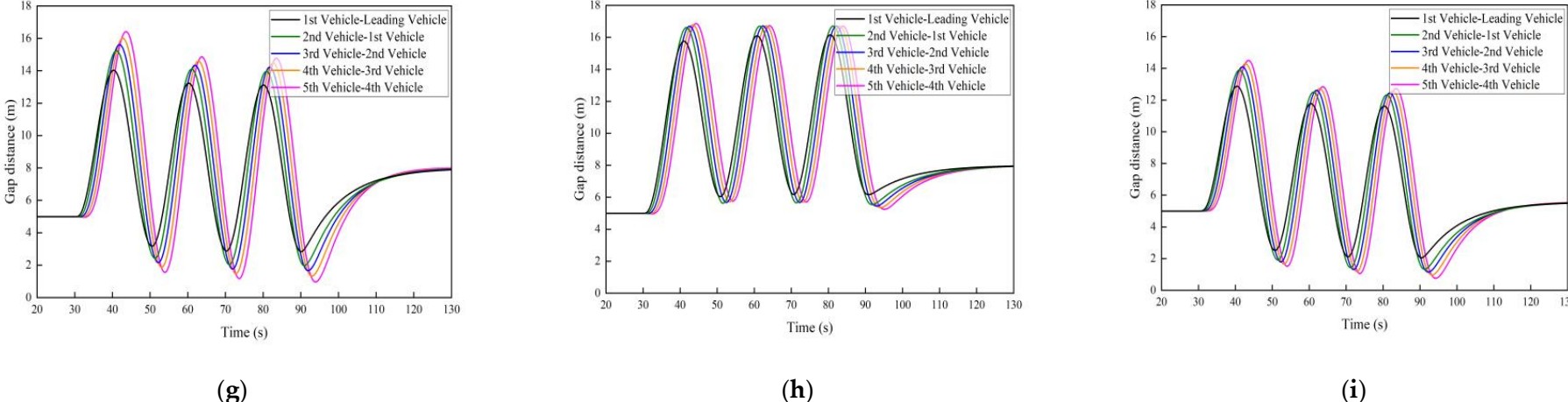

(**g**)　　　　　　　　　　　　　　　　(**h**)　　　　　　　　　　　　　　　　(**i**)

**Figure 11.** Comparison results under three different control strategies. (**a**) Velocity changes in the constant spacing strategy; (**b**) Velocity changes in the constant time headway strategy; (**c**) Velocity changes in in the potential field-based strategy; (**d**) Acceleration changes in the constant spacing strategy; (**e**) Acceleration changes in the constant time headway strategy; (**f**) Acceleration in the potential field-based strategy; (**g**) Gap distance changes in the constant spacing strategy; (**h**) Gap distance changes in the constant time headway strategy; (**i**) Gap distance changes in the potential field-based strategy.

Figure 11g–i depict the change in the gap distance between each adjacent vehicle under three different control strategies. It can be found that in the constant time headway strategy, the gap distance value is generally higher than the other strategies. This is because in this strategy, the gap distance will increase as the velocity increases. This control strategy is to ensure the speed and acceleration tracking effect of the vehicle with a large traffic efficiency sacrifice. In contrast, we can find that the control optimization strategy based on the G-SPF model proposed in this paper shows obvious advantages in terms of improving traffic efficiency.

In order to compare the three strategies more intuitively, we summarize the comprehensive performance of the platoon in terms of efficiency, safety and comfort under different strategies, as shown in Table 3.

**Table 3.** Comprehensive performance for different control strategies.

| Performance Indicators | Control Strategies | | |
|---|---|---|---|
| | Constant Spacing Strategy | Constant Time Headway Strategy | Potential Field-Based Strategy |
| Average time headway (s/veh) | 0.839 | 1.011 | 0.698 |
| Travel time (s) | 110.7 | 113.5 | 108.6 |
| Velocity deviation (m/s) | 0.071 | 0.057 | 0.013 |
| Average jerk $(m/s^3)$ | 0.215 | 0.193 | 0.202 |

The average time headway (s/veh) represents the average headway between adjacent vehicles in the platoon that is obtained by calculating the headway based on the vehicle gap distance and velocity at each moment. Compared with the constant spacing and constant time headway control strategies, the simulation results show that our proposed method can strengthen the traffic capacity by approximately 48.8% and 26.6%, respectively. The travel time (s) refers to the time when all vehicles in the platoon passed the section (in this paper, we set the end point of the section to a position of 1000 m in the lateral direction). These two indicators mainly reflect the efficiency of the platoon. The velocity deviation (m/s) represents the average value of the corresponding velocity deviation between all following vehicles and the leading vehicle. A larger velocity deviation always means worse safety. The average jerk $(m/s^3)$ refers to the change rate in vehicle acceleration, which represents the driving comfort degree during the driving process. A smaller average jerk indicates higher driving comfort. We can find that the constant time headway control method has the highest driving comfort, but it is based on the premise of sacrificing a lot of traffic efficiency.

On the whole, the G-SPF-based strategy has the best overall performance, especially in terms of improving traffic efficiency. Note that when the velocity of the platoon is higher, the constant time headway strategy will produce a much larger gap distance, which will be unfavorable for traffic efficiency.

## 5. Conclusions

This study proposes a novel graph and safety potential field theory-based platoon formation and optimization method. Graph theory is applied to describe the information topology between CAVs and control the trajectory of CAVs during the platoon formation process. Safety potential field theory is exploited to characterize the driving risk of CAVs during the platoon formation process. A four-step platoon formation and optimization control strategy based on the G-SPF model is proposed. This platoon control method increases the consideration of the complex driving environment during the formation of the platoon, which ensures the driving safety of vehicles in the CAVs environment. CAVs distributed in different lanes can be aggregated into the lane where the leading vehicle is located according to the information between the CAVs without any collisions. The

optimized control after the formation of the platoon shows obvious advantages in traffic efficiency and exhibits strong platoon stability, compared with the traditional fixed spacing or fixed headway control method.

From the perspective of platoon configurations in the future connected and automated environment, safe and efficient platooning has always been the research direction we are pursuing. The findings of this study illustrate that the safety potential field model, considering the multiple pieces of motion information of vehicles (velocity, acceleration, steering angle, etc.), can be well applied to vehicle platoon control. There are some limitations to this study that would need further improvements. More available field data need to be collected for the purpose of calibration and validation of our proposed model. In addition, there is no discussion of platoon formation under different vehicle information topologies. Therefore, further studies may focus on the analysis of multi-type topologies based on real vehicle experiments and try to combine different types of advanced control methods to further expand the research of this study. Nevertheless, this paper provides a new idea and method for platoon formation and optimization in the future autonomous driving environment.

**Author Contributions:** Conceptualization, L.L. and J.G.; methodology, L.L.; software, P.M.; validation, J.G., Z.Y. and X.Q.; formal analysis, L.L.; writing—original draft preparation, L.L. and J.G.; writing—review and editing, X.Q.; visualization, L.L., P.M. and Z.Y.; supervision, X.Q. and B.R.; project administration, X.Q.; funding acquisition, X.Q. All authors have read and agreed to the published version of the manuscript.

**Funding:** This research was supported by the National Key R&D Program in China (Grant No. 2018YFB1600600), the MOE (Ministry of Education in China) Project of Humanities and Social Sciences (Project No. 20YJAZH083) and the National Natural Science Foundation of China (Grant No.51878161). Part of the research was conducted at the University of Wisconsin-Madison where the first author spent a year as a visiting student funded by the China Scholarship Council.

**Informed Consent Statement:** Informed consent was obtained from all subjects involved in the study.

**Data Availability Statement:** Data sharing not applicable. No new data were created or analyzed in this study. Data sharing is not applicable to this article.

**Acknowledgments:** We would like to thank the anonymous reviewers whose insightful comments have helped significantly improve the quality of the analysis and presentation of this study. Part of the research was conducted at the University of Wisconsin-Madison where the first author spent a year as a visiting student funded by the China Scholarship Council.

**Conflicts of Interest:** The authors declare no conflict of interest.

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
