# Peer review of "A Novel Graph and Safety Potential Field Theory-Based Vehicle Platoon Formation and Optimization Method"

_applsci, doi:10.3390/app11030958_

Round 1
Reviewer 1 Report
The issues presented in the article are very interesting, but after reading the text carefully, the following issues appeared:
- Table 1: it seems that part of the text of the table headings is unreadable - obscured;
- line 144: the left side of the equation "... = (V, E)" is missing;
- formulas (9), (10), (11), (12), (13), (14), (15), (16): explanation of all variables is missing - especially in the context of the model discussed in the article; Some of the variables and coefficients would be useful to highlight in Fig. 1, Fig. 2, Fig. 3, and Fig. 4; Part of the description is only on page 10;
- the paper has the same markings for different variables, eg V for Velocity (line 310) and for Vertex (line 145);
- Fig.4.: in flowchart it is not known what the "t" variable is for; What event / variable / variable value causes the algorithm to terminate (STOP)?
- Sections 3 or 4: the description of the tool and the computational technique in which the calculations were performed are missing;
- Fig. 6, Fig.8, Fig.10, Fig.11: it is recommended to enlarge the charts so that they have better readability;
- there is no comparison of the obtained results with other models.
Reviewer 2 Report
The article concerned a very interesting scientific issue, which is the concept of platoning. Authors noticed that Platooning is considered to be a very effective method for improving traffic efficiency, traffic safety and fuel economy under connected and automated environment, and I totally agree with that.
The authors did a lot of work to prepare the research and the article is very veluable.
I sugest to prepare two corrections. First is to improve the article accoriding to thr MDPI template. First of all I think that the literature review chapter is very short and focused mostly on the articles prapred by the Chinese researchers. I suggest to improve that part and add more literature references:
- https://www.mdpi.com/2071-1050/10/7/2404
- https://doi.org/10.3390/app10165580
- https://doi.org/10.3390/en13215810
- https://doi.org/10.3390/su12218841
- https://doi.org/10.20858/sjsutst.2018.100.2
Also think it will be very useful to add the "discussion" chapter and compare the results with the articles prepared by the other researchers. Maybe some table comparison of the results?
After these two amendments are prepared, the article will be ready for publication
Reviewer 3 Report
This paper deals with the problem of platooning vehicles in an automated and connected environment. A new graph and safety potential field theory-based approach is proposed and verified in numerical experiments. A comparison with two existing approaches is made to prove the effectiveness of the new approach. The good point of the paper is a good description of existing approaches and the proposed approach. The main drawback of the paper is related to the creation of the first scenario for numerical evaluation. Initial spatial distribution of the vehicles has to be more realistic, or a more detailed description of the scenario must be provided to explain the motivation for this spatial vehicle distribution.
In line 113, you mention complex traffic environment factors, but there is no explanation of these factors. More information would be welcome.
In line 205, you refer to two vehicles connected by a mutual communication connection? Rephrase to make it more understandable.
In line 340, you are referring to a target vehicle. There is no explanation of the target vehicle's characteristics or which vehicle in the platoon is the target vehicle. Please elaborate on this.
Fig. 5 presents the vehicles' spatial distribution on a three-lane road/motorway if I understand everything correctly. Good vehicle starting positions should be in the middle of the respective lane. Oddly, some vehicles have a starting position on the horizontal road markings. A comment about the starting vehicle positions for the made numerical experiments would be welcome. The experiments should be realistic, and drivers usually place their vehicles in the middle of a lane. Scenario II gives more realistic starting positions in that context.
In the paragraph between lines 437 and 445 emphasize that the additional simulations with the disturbance are combined with two existing platooning approaches. It is unclear what results are gathered in Fig. 11.
In line 469, emphasize the parameters that were improved regarding traffic efficiency.
Add punctuation to equations if they are part of a sentence. Also, enumerate all equations. All variables used in equations have to be explained in the text.
Some improvement of the language would be welcome.
There are additional comments in the attached PDF.

Reviewer 4 Report
An excellent organized and written manuscript which presents a new idea and method for the platoon formation and optimization in the future autonomous driving environment.
Author Response
Thank you very much for your affirmation of our work in this paper. We appreciate your time and help in reviewing our manuscript and the insightful comments you provided that have helped significantly improve the quality of this study.
Round 2
Reviewer 1 Report
The additions and corrections introduced by the authors are satisfactory.
Author Response

(The authors gave the same response as above.)

Reviewer 2 Report
The article is corrected and I accept it in present form.
Author Response

(The authors gave the same response as above.)

Reviewer 3 Report
It can be seen that the paper has been improved. All major drawbacks have been addressed. There are some minor issues and typos that I have denoted in the attached PDF. Pay attention to the fact that if an equation is part of a sentence, you add punctuation (comma if the sentence continues after the equation or point if the equation ends the sentence).

Author Response
Thank you very much for the detailed comments. We appreciate your time and help in reviewing our manuscript. We have revised the paper very carefully according to your suggestion. All the modified contents have been marked in red font in the updated manuscript.